# MWCNT Decorated Rich N-Doped Porous Carbon with Tunable Porosity for CO_2_ Capture

**DOI:** 10.3390/molecules26113451

**Published:** 2021-06-07

**Authors:** Yuanjie Xiong, Yuan Wang, Housheng Jiang, Shaojun Yuan

**Affiliations:** Low-Carbon Technology & Chemical Reaction Engineering Lab, College of Chemical Engineering, Sichuan University, Chengdu 610065, China; 2018223075141@stu.scu.edu.cn (Y.X.); wangyuan592675365@163.com (Y.W.); 2019323070018@stu.scu.edu.cn (H.J.)

**Keywords:** polyaniline, porous carbon, nitrogen doping, carbon nanotube, CO_2_ capture

## Abstract

Designing of porous carbon system for CO_2_ uptake has attracted a plenty of interest due to the ever-increasing concerns about climate change and global warming. Herein, a novel N rich porous carbon is prepared by in-situ chemical oxidation polyaniline (PANI) on a surface of multi-walled carbon nanotubes (MWCNTs), and then activated with KOH. The porosity of such carbon materials can be tuned by rational introduction of MWCNTs, adjusting the amount of KOH, and controlling the pyrolysis temperature. The obtained M/P-0.1-600-2 adsorbent possesses a high surface area of 1017 m^2^ g^−1^ and a high N content of 3.11 at%. Such M/P-0.1-600-2 adsorbent delivers an enhanced CO_2_ capture capability of 2.63 mmol g^−1^ at 298.15 K and five bars, which is 14 times higher than that of pristine MWCNTs (0.18 mmol g^−1^). In addition, such M/P-0.1-600-2 adsorbent performs with a good stability, with almost no decay in a successive five adsorption-desorption cycles.

## 1. Introduction

Anthropogenic carbon dioxide (CO_2_) emission from the fuel fossil combustion causes a series of ecological crises, including global warming, glacial melting, and species numbers dropping [1,2]. The raised temperature of land also slows down the airflow between land and sea, further causing the fog and haze in urban areas. The alarming issue of the continual rise in CO_2_ concentration is a serious concern to the public, and plenty of strategies were urged to mitigate the greenhouse gas in various industries. Carbon capture and storage (CCS) was considered as an efficient approach to address the challenge of excessive CO_2_ emissions [3,4]. It is well known that amine solutions could effectively absorb CO_2_, however the high cost, corrosion, and environmental pollution limits its practical application [5,6]. The adsorption technologies of temperature swing adsorption (TSA) and vacuum swing adsorption (VSA) are efficient strategies for gas storage and separation, suggesting that solid adsorbents are the most promising candidates to adsorb CO_2_ [7,8]. The mechanism of CO_2_ uptake on carbonaceous sorbents was ascribed to the porosity and surface chemistry properties, which provide physical and chemical adsorption of CO_2_, respectively. Therefore, numerous porous adsorbents have been developed, including porous carbon, zeolites, porous polymer, and N-doped carbon [9,10,11,12].

Carbon nanotubes (CNTs) have drawn much attention for CO_2_ uptake due to their unique physicochemical properties and high thermal and chemical stability [13,14]. The developed porosity provided more possibilities for grafting an amine functional group, which can further improve the chemical adsorption of CO_2_. Keren et al. reported that impregnating 3-aminopropyl-triethoxysilane (APTS) onto CNTs increased the CO_2_ adsorption capacity of the CNTs by 150% [15]. Lee et al. employed polyethyleneimine to treat multi-walled carbon nanotubes (MWCNTs), and a 200% increase in CO_2_ adsorption capacity was achieved [16]. In addition, P-phenylenediamine grafted onto MWCNTs showed a 250% improvement of CO_2_ adsorption, and the CO_2_ adsorption capacity reached 0.59 mmol g^−1^ at 303.15 K and 200 kPa [17]. Recently, much attention was focused on N-doped porous carbons (NPCs), due to their large surface area and dense active nitrogen with negative charge, which can offer more active sites for CO_2_ uptake [18,19,20].

The fabrication of NPCs is often through the post-treatment of carbon with an inorganic nitrogen resource, such as urea or ammonia at high temperatures and the pyrolysis of N containing organic monomers. Polyaniline (PANI) is a rich N-containing polymer, and it is usually employed to be a coating on the substrate, which is promising precursor to fabricate NPCs [21,22]. Zhang et al. investigated the activation process of PANI derived NPCs by using KOH as the activator, and achieved a high surface area NPC with a surface area of 3768 m^2^ g^−^^1^ [23]. Silvestre-Albero et al. obtained a series of NPCs by the carbonization of PANI at different temperatures [24]. The high surface area and high N content made such NPCs deliver a good CO_2_ capture capability. In addition, Khalili et al. prepared N-doped activated carbon/PANI, and it delivered an enhanced CO_2_ adsorption capability and selectivity than that of pure activated carbon [25]. We anticipate that the NPCs derived from the PANI coating on the surface of CNTs can provide more active sites, which favor CO_2_ adsorption more energetically. Furthermore, the framework provided by CNTs also supports NPCs to construct the ordered channels for transferring CO_2_ molecules. To the best of our knowledge, however, few reports have explored this to date.

In this work, a novel adsorbent of MWCNTs/NPCs was fabricated by the in-situ chemical oxidation polymerization of PANI on a surface of MWCNTs, and then activated with KOH (see experiment section and Scheme 1 for detailed preparation). The porosity of such NPCs can be tuned by the rational addition of MWCNTs, and adjusting the amount of KOH and the pyrolysis temperature. The obtained M/P-0.1-600-2 adsorbent, with a high surface area of 1017 m^2^ g^−1^ and a high N content of 3.11 at%, possessed a boosted CO_2_ adsorption capacity of 2.63 mmol g^−1^, which is higher than that of pristine MWCNTs (0.18 mmol g^−1^) and pure PANI (0.54 mmol g^−1^). Moreover, such M/P-0.1-600-2 adsorbent performed with a good stability, with almost no decay in a successive five adsorption-desorption cycles.

## 2. Results and Discussion

As shown in Scheme 1, PANI was first prepared by in-situ oxidative polymerization on the surface of MWCNTs. The obtained MWCNTs/PANI nanocomposite was denoted as M/P-x, where the “x” represents the mass ratio of MWCNTs and aniline monomer. Then, the M/P-x was further activated at different temperatures with KOH addition. The products were named as M/P-x-T-y, where the T is the activation temperature, and the y is the mass ratio of KOH and M/P (see experiment section for detailed preparation). Figure 1a shows the Fourier transform infrared (FTIR) spectra of M/P-0.1-600-2, M/P-0.1, PANI, and MWCNT samples. MWCNTs display three peaks at 1402 cm^−1^, 1635 cm^−1^ and 3437 cm^−1^, corresponding to stretching vibration of C=C, -OH, and the bending vibration of C=O, respectively [26]. This indicates the successful pre-oxidation reaction of MWCNTs. The PANI shows the peaks at 803 and 1289 cm^−1^, which is ascribed to the stretching of C-C and C-N, respectively. The peaks at 1243 and 1567 cm^−1^ are attributed to C-NH^+^ and stretching vibration quinoid rings [27]. After coating PANI on the surface of MWCNTs, the FTIR spectrum of M/P-0.1 shows both the MWCNTs and PANI characteristic peaks. After activation, the characteristic peaks of PANI disappeared, which is ascribed to successful carbonization of PANI to form the N-doped porous carbon. In addition, the FTIR spectra of M/P-x (x = 0, 0.03, 0.05, 0.1, 0.2, and 0.3), M/P-0.1-T-2 (T = 0, 300, 400, 500, 600, and 700), and M/P-0.1-600-y (y = 0, 1, 2, and 4) are also illustrated in Appendix A, respectively. X-ray diffraction (XRD) patterns of four samples are shown in Figure 1b. Notably, M/P-0.1 shows the similar characteristic peaks of PANI, implying the existence of PANI on MWCNTs [28,29]. After activation, the characteristic peak at 42.9° is observed corresponding to the MWCNTs, which is ascribed to the break of porous carbon coating after the activation reaction. This result further confirms the PANI encapsulated the surface of MWCNTs by the in-situ chemical oxidation reaction. Appendix A also display the XRD patterns of the precursor M/P-x, M/P-0.1-T, and M/P-0.1-600-y, respectively. Raman spectra of four samples (Figure 1c) show the two broad peaks at 1355 and 1584, which are attributed to D and G bands, respectively. Compared to M/P-0.1, the Raman spectrum of M/P-0.1-600-2 shows a decreased ratio of G band to D band (I_G_/I_D_), implying more defects after activation reaction. This can provide more active surface favoring the transfer and adsorption of CO_2_ molecule. Thermogravimetric analysis (TGA) curves of MWCNTs (Figure 1d) show a good thermal stability, with only 3.24% weight loss. However, four obvious mass loss stages can be observed for pure PANI at the range of 0–120, 250–300, 300–480, and 480–600 °C, corresponding to the evaporation of free water, the decomposition of dopant and oligomer, the decomposition of amine groups on PANI, and the decomposition macromolecular benzene ring and quinonoid ring, respectively [30]. Notably, M/P-0.1-600-2 exhibited an enhanced thermal stability. In addition, the TGA curves of all samples fabricated at different MWCNTs and KOH addition, and activation temperatures are shown in Appendix A.

The morphology of nanotubes cannot be observed in the scanning electron microscopy (SEM) images of M/P-0.1 (Figure 2a–c), which is due to the MWCNTs intertwined with PANI coating. After activation, a relatively rough surface is observed (Figure 2d–f), indicating the formation of porous carbon, which is beneficial to the CO_2_ adsorption. Transmission electron microscopy (TEM) images of M/P-0.1 (Figure 2g–j) displays the nanotube morphology with a uniform PANI coating, which further indicates the effective in-situ chemical oxidation reaction. Hence, the formation of N-doped porous carbon after activation can uniformly disperse on the surface of MWCNTs. As shown in Figure 2j–l, M/P-0.1-600-2 shows the clear nanotube morphology, and the larger interface of each nanotube can be observed, favoring the transfer of CO_2_.

To investigate the surface chemistry, X-ray photoelectron spectroscope (XPS) measurement was employed. Figure 3 and Appendix A compare the XPS spectra of M/P-0.1-T-2 prepared in different activation temperatures. The wide scan XPS spectra (Figure 3a,d and Appendix A) show the existence of C, N, and O elements on surface of all samples. The N content of each sample is listed in Appendix A. The N content of M/P-0.1 is measured to be 4.14%. After activation, the N content of 3.11% is still measured. It is of note that the N content decreased with the raised pyrolysis temperature, which is due to the instability of active N in the carbon framework. As for C 1s (Figure 3b,e and Appendix A), four peaks at around 284.5, 285.2, 286.3, and 288.4 eV are observed indicating the C-C/C=C, C-N/C-O, C=O, and O=C-O, respectively [31]. Figure 3c,f and Appendix A compare the N 1s core-level XPS spectra of all samples. Two peaks with binding energy of 398.4 and 400.3 eV are ascribed to the pyridine-N and pyrrolic-N, respectively [32]. Notably, more pyrrolic-N is observed in the M/P-0.1-600-2, which more favors the CO_2_ capture [33,34].

A relatively larger generation of pyrrolic-N at the activation temperature of 600 °C could be attributed to the increased defects, facilitating the exposure of active nitrogen in carbon framework, which can be further confirmed by Brunauer–Emmett–Teller (BET) measurement. N_2_ adsorption-desorption isotherms of four samples are shown in Figure 4a. Different from the shape of MWCNTs, M/P-0.1, and PANI, M/P-0.1-600-2 exhibits a sharply increased adsorption quantity at *P*/*P*_0_ < 0.03, implying type I (IUPAC classification) property for microporous materials [35]. Furthermore, N_2_ adsorption-desorption isotherm of M/P-0.1-600-2 shows an obvious hysteresis hoop at *P*/*P*_0_ = 0.4–0.7, indicating the existence both micro- and mesopore feature [36]. In addition, the isotherms of MWCNTs and M/P-0.1 displayed a sharply increased hysteresis loop at *P*/*P*_0_ =0.9–1.0, indicating a mesopore structure. The corresponding pore size distribution of MWCNTs (Figure 4b) exhibits the mesopore and macropore feature with the range of 25–80 nm. After combination with N-doped carbon, M/P-0.1-600-2 mainly presents the micro- and mesopore nature. To further investigate the porosity, the N_2_ adsorption-desorption isotherms and corresponding pore size distribution of all samples at different prepared conditions are displayed in Appendix A, respectively, and the detailed parameters are listed in Appendix A. Clearly, before activation reaction, the addition of MWCNTs can facilitate a larger surface area (Appendix A). However, for the activation samples of M/P-x-600-2, the surface area is limited by MWCNTs introduction (Appendix A). Possibly, this result could be ascribed to the superior porosity of PANI after carbonization and activation. With the increased temperature from 300 to 600 °C, the increased surface area of M/P-0.1-T-2 is observed. The surface area of M/P-0.1-600-2 is calculated to be 1017 cm^2^ g^−1^. This value is higher than that of M/P-0.1-700-2 (780 cm^2^ g^−1^), which is due to the overactivation under a high temperature, leading to the destruction of the pore structure. This result can be further confirmed by the considerably increased pore volume from 0.37 for M/P-0.1-600-2, to 0.94 for M/P-0.1-700-2 (Appendix A). In addition, the dosage of KOH is also a significant factor for porous carbon preparation. Excessive KOH addition also leads to an overactivation, limiting the development of porosity (Appendix A).

Figure 5a exhibits the adsorption isotherms of MWCNTs and M/P-0.1-600-2 at different pressure and 298.15 K. Notably, owing to the large surface area and high N content, M/P-0.1-600-2 displays a superior CO_2_ adsorption capacity of 2.63 mmol g^−1^, which is about 14 times higher than that of MWCNTs (0.18 mmol g^−1^) and PANI based NPCs (1.16 mmol g^−^^1^). The detailed comparison is listed in Appendix A. Notably, it seems that the surface area is significant for the CO_2_ adsorption capacity. However, M/P-0.03-600-2 (1570 cm^2^ g^−1^) and M/P-0.05-600-2 (1601 cm^2^ g^−1^) possessing a high surface area only delivers a relatively poor CO_2_ capture capability with 1.25 and 1.63 mmol g^−1^, respectively. Possibly, the introduction of MWCNT constructs results in more pathways for CO_2_ adsorption, as M/P-0.1-600-2 presents relatively large pore volume (0.37 cm^3^ g^−1^) and smaller average pore size (2.71 nm). In addition, more comparisons of other porous adsorbent for CO_2_ capture are listed in Appendix A. In order to further investigate the relationship between M/P-0.1-600-2 and CO_2_, three models (i.e., Redlich–Peterson, Freundlich and Langmuir) were employed [37], as illustrated in Figure 5b and Appendix A, and the corresponding parameters are listed in Appendix A. The data of the Redlich–Peterson model presents a higher value of R^2^, implying the monolayer and multi-layer adsorption process between the M/P-0.1-600-2 and CO_2_. In addition, Fick models and linear driving force (LDF) [38] were also conducted to evaluate the adsorption kinetics of M/P-0.1-600-2, as shown in Figure 5c. The fitted parameters (Appendix A) shows that Fick models possesses a higher R^2^ and lower χ^2^ coefficient value than that of LDF, indicating that the CO_2_ adsorption kinetics of M/P-0.1-600-2 follows the Fick model, suggesting that the physical adsorption plays a major role in the adsorption process [39]. The regeneration property is significant for solid adsorbents. After saturated with CO_2_ at 298 K and five bars, the adsorbent is regenerated in a vacuum drying chamber for approximately 40 Pa at 373.15 K, and then re-employed for CO_2_ adsorption. Notably, after a successive five recycling tests, a CO_2_ adsorption capacity retention of 96.7% still remained (Figure 5d), suggesting the outstanding recycling stability for such M/P-0.1-600-2 absorbent.

## 3. Materials and Methods

### 3.1. Materials

Aniline monomer (99.8%) is an analytical grade and purchased from Sigma-Aldrich Co. (Shanghai, China). MWCNTs (98%) and ethanol (AR, 98%) were obtained from Hengqiu Technology Reagent Co. (Suzhou, China) and Changzheng Chemical Reagent Co. (Chengdu, China). Ammonium persulfate, potassium hydroxide, and hydrochloric acid are all analytically pure grades, and purchased from Kelong Chemical Reagent Co. (Chengdu, China).

### 3.2. Fabrication of MWCNTs/PANI Nanocomposite

A total of 0.5 g MWCNTs were added to 40 mL mixed acid, with a volume ratio of sulfuric acid and nitric acid of 3:1, and the mixture was stirred at 90 °C for 90 min. After cooling to room temperature, the mixture was washed by deionized water. The carboxylation of the MWCNTs was obtained by drying in a vacuum drying oven at 60 °C for 12 h, and then ground into powder. A total of 0.2 g (2.144 mmol) aniline monomer was dissolved into 15 mL 1 M HCl solution, then a certain amount of the carboxylated MWCNTs were dispersed into the solution with string for 60 min in an ice/water bath to obtain “solution A”. Then, 0.334 g (1.465 mmol) ammonium persulfate (APS) was dissolved into 5 mL 1 M HCl solution by stirring for 30 min in an ice/water bath to obtain “solution B”. Then, the precooled (0 °C) solution B was dropped into solution within 30 min, and then allowed to react for 6 h in ice bath. Finally, the product solution was filtrated with a Brucella funnel and washed with deionized water and ethanol until the supernatant turned transparent and pH turned neutral. The compound was frozen to ice in liquid nitrogen, then dried by freeze dryer for 24 h. The MWCNTs/PANI nanocomposites with different mass ratios of MWCNTs and aniline monomer (0.3:1, 0.2:1, 0.1:1, 0.05:1, and 0.03:1) were obtained by adjusting the content of MWCNTs. For convenience, the MWCNTs/PANI nanocomposite was denoted as M/P-x, where the “x” represents the mass ratio of MWCNTs and aniline monomer. For example, M/P-0.1 represents the composite with the mass ratio of MWCNTs and aniline of 0.1. For comparison, the neat PANI was synthesized under the same conditions without the addition of MWCNTs.

### 3.3. Activation of MWCNTs/PANI Nanocomposite

The well mixed M/P nanocomposites and KOH activator were preheated under N_2_ flow (60 mL min^−^^1^) at 200 °C for 1 h in a tubefurnace, then heated to the specified temperature at a rate of 5 °C min^−1^, left to activate for 2 h, and finally the activated sample was washed by ethanol and deionized water until the pH value of filtrate was approzimately7. The product was then obtained by drying in vacuum oven at 80 °C for 12 h. Different samples were fabricated under different KOH mass ratios (KOH:M/P = 1, 2, and 4) of the M/P nanocomposite and the activator, and different activation temperatures (300, 400, 500, 600, and 700 °C). The final products were named as M/P-x-T-y, where the T is the activation temperature, and the y is the mass ratio of KOH and M/P. For example, M/P-0.1-600-2 means that the mass ratio of KOH activator and M/P-0.1 is 2, and the activation temperature is 600 °C.

### 3.4. Characterization

The morphology of the pristine and activated composites was observed by the SEM (JSM-5900LV, JEOL, Tokyo, Japan). The FTIR spectra were obtained by a Spectrum GX FTIR spectrometer (Perkin15 Elmer Inc., Waltham, MA, USA) to identify chemical functional groups on the sample surface, operating between 4000 and 400 cm^−1^ with a resolution of 4 cm^−1^, using the KBr tablet method. The crystal structures of all samples were determined by using XRD (DX2700, Haoyuan Instruments Co., Dandong, China) with Cu Kα radiation (k = 1.5418 Å) produced at 40 kV and 40 mA. XRD patterns were recorded at a diffraction angle range of 2θ = 5–90° under a continuous scan using 0.06° step size. TGA was fulfilled with a thermal graphic analyzer (HTG-2, Beijing Hengjiu Science In-strument Factory, Beijing, China) with a heating rate of 10 K min^−^^1^ from room temperature to 1073 K under N_2_ flow (60 mL min^−1^). TEM (Tecnai G2 F20S-TWIN, FEI, Hillsboro, OH, USA) was used to observe the thickness changes of the samples before and after coating. The surface compositions of M-P nanocomposite fabricated under different conditions were characterized by XPS (AXIS NOVA, Kratos, UK). The pore structure characteristics of the porous materials were determined by nitrogen adsorption at −196 °C using TristarII3020 analyzers (Micromeritics, Norcross, GA, USA), and all samples were degassed at 150 °C under vacuum at 200 Pa for a period of at least 12 h. The surface area was calculated using the multipoint Brunauer–Emmett–Teller (BET) method. Raman spectra were recorded with a DXR Raman Microscope (Thermo Scientific, Waltham, MA, USA) using an excitation wavelength of 514.5 nm.

### 3.5. CO_2_ Adsorption Experiment

CO_2_ adsorption capacity was measured by self-made static adsorption apparatus (see the Appendix A for detailed ascribed).

## 4. Conclusions

In summary, we demonstrated a novel method to prepare a MWCNT decorated N-doped porous carbon. The porosity of the absorbent can be well controlled by the rational addition of MWCNTs, tuning the pyrolysis temperature, and adjusting the amount of KOH. One such result, M/P-0.1-600-2, possessed a high surface area (1017 m^2^ g^−1^) and high N content (3.11 at%), and was an efficient adsorbent, presenting a good CO_2_ adsorption capacity of 2.63 mmol g^−1^ at 298.15 K and five bars. Notably, it also exhibited a good recycling stability in a successive five adsorption-desorption tests.

## Data Availability

Data are contained within the article.

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
