# Peer review of "MWCNT Decorated Rich N-Doped Porous Carbon with Tunable Porosity for CO2 Capture"

_molecules, 2021, doi:10.3390/molecules26113451_

Round 1

Reviewer 1 Report

1. The authors should add more information on why carbon dioxide should be removed from the environment in the introduction section.

2. According to the SI standards, there should be no space between the value and Celsius degree.

3. Please add a citation as it will improve the manuscript: https://doi.org/10.3390/ma13122782

4.Authors should get spell-checked manuscript by a native speaker as there are numerous grammar and spelling mistakes ex. Line 179 - Langmuir rather than Langmiur.

Author Response

Reviewers:

  1. The authors should add more information on why carbon dioxide should be removed from the environment in the introduction section.

Response

As suggested, we added the related information in the introduction section (See page 1).

Reviewers:

  1. According to the SI standards, there should be no space between the value and Celsius degree.

Response

We appreciate for reviewer’s comment. As suggested, the mistake has been corrected. Furthermore, we also review the whole manuscript carefully.

Reviewers:

  1. Please add a citation as it will improve the manuscript: https://doi.org/10.3390/ma13122782

Response

As suggested, the citation has been added (Ref.38).

Reviewers:

  1. Authors should get spell-checked manuscript by a native speaker as there are numerous grammar and spelling mistakes ex. Line 179 - Langmuir rather than Langmiur.

Response

We appreciate for reviewer’s comment. As suggested, the mistake has been corrected.

Reviewer 2 Report

The authors present a study about CO2 sorbents made from MWCNTs and PANI composites activated in KOH. KOH activation to prepare porous N-doped carbon materials from PANI is known, please see two examples https://doi.org/10.1039/C5RA13515J & https://doi.org/10.1021/ie5013129. A number of experimental techniques were used to characterise the activated materials, for this reason, the work is valuable in the extent of data presented. There are a number of main issues to address before further consideration.   - Written English needs to be improved to the clarity, coherence, and correctness essential to understand and appreciate the value of the study.   - KOH activation of other PANI and PANI-based composites need to be reviewed thoroughly adding all relevant references in the paper. The authors should demonstrate how novel their materials are with the many already developed following common alkali activation approaches.   3) The CO2 sorption performance of M/P-0.1-600-2,  2.63 mmol g-1 at 298.15K and 5 bar, should be compared to that of other porous carbons. Please add a table (in addition to Table S2 where only materials specific to this study are listed) to compare to those developed by other research groups.   4) Abbreviations of materials are not described at the start of the paper, only later in Section 4. While this is in line with the template of the journal, the reader cannot follow the discussion in Section 2 since the abbreviations are not explained. More about this issue is described below.  

The manuscript is awkward to read due to the use of unsuitable grammar or sentence structure, or lack of meaning, for example:

"Although the great improvement of CO2 adsorption was achieved by amine-functionalized CNTs, however, it cannot fulfill [sic.] the ever-increasing demand of the human development."   What does it mean that the amine-functionalized CNTs achieved the "importance" of CO2 adsorption? How can a material (in this case amine-functionalized CNTs) fulfil the demand (of what?) of human development? It is difficult to capture the meaning of this sentence.    Five adsorption/desorption cycles are very few for sorbent materials that must sustain thousands of cycles in industrial applications, for this reason, the statement "outstanding stability with almost no decay in a successive five adsorption-desorption cycles" is misleading and wrongly placed.   Section 2, Results and Discussion: This section immediately starts with data analysis without introducing and explaining the meaning of sample name abbreviations, the reader is left wondering what  M/P-0.1-600-2 and M/P-0.1 are, for example. Sample names/abbreviations must be explained before discussing the results otherwise it is not possible to follow the paper. Note that the reader cannot wait to reach Sections 4.2 and 4.2 to finally find the required explanations. It appears that the authors moved the section "Materials and Methods" to the end of the manuscript, as required by the journal template, without assuring that the paper could be understood by the reader. Also, immediately after, the authors write about the "successful peroxidation reaction of MWCNTs", but this reaction is not described, again the reader is left with no information about how the materials were prepared. As a matter of fact, the word "peroxidation" is mentioned only once in the paper, so there is either no information about the reaction or it is not possible to find it. Later on (line 84) another sample abbreviation is introduced "M/P-0.1-T-2" but again without any previous description of what that is and how it was prepared.   Please revise this statement:   "The fitted parameters (Table S4) shows [sic.] that LDF models possesses [sic.] a higher R2 and lower χ2 coefficient value than that of Fick, indicating that the CO2 adsorption kinetics of M/P-0.1-600-2 follows the LDF model, suggesting fast initial physical adsorption. The regeneration property is the [sic.] significant for solid adsorbents."   In Table S4 the R2 of the Fick model is better than that of LDF, the same applies for χ2. Also, the LDF fitting line does not follow the experimental data in Figure 5c, so it is not correct to state that "the CO2 adsorption kinetics of M/P-0.1-600-2 follows the LDF model".   The wording "raman spectrum" is incorrect since Raman must be capitalized.   Line 215: "Then, The" double capitalized sentence.   Please proofread very carefully to remove all English issues.

Author Response

Reviewers:

The authors present a study about CO2 sorbents made from MWCNTs and PANI composites activated in KOH. KOH activation to prepare porous N-doped carbon materials from PANI is known, please see two examples https://doi.org/10.1039/C5RA13515J & https://doi.org/10.1021/ie5013129. A number of experimental techniques were used to characterise the activated materials, for this reason, the work is valuable in the extent of data presented. There are a number of main issues to address before further consideration.   

Response

We appreciate for the reviewer’s insightful comments. The manuscript has been revised carefully point-by-point accordingly.

Reviewers:

(1) Written English needs to be improved to the clarity, coherence, and correctness essential to understand and appreciate the value of the study.   

Response

As suggested, we carefully review the whole manuscript and modify these mistakes.

Reviewers:

(2) KOH activation of other PANI and PANI-based composites need to be reviewed thoroughly adding all relevant references in the paper. The authors should demonstrate how novel their materials are with the many already developed following common alkali activation approaches.

Response

As suggested, we added the relevant references in introduction section (See pages 2).

Reviewers:

(3) The CO2 sorption performance of M/P-0.1-600-2, 2.63 mmol g-1 at 298.15K and 5 bar, should be compared to that of other porous carbons. Please add a table (in addition to Table S2 where only materials specific to this study are listed) to compare to those developed by other research groups.

Response

As suggested, the comparison was added as shown in Table S3 (Ref.1–Ref.9 in supporting information).

Reviewers:

(4) Abbreviations of materials are not described at the start of the paper, only later in Section 4. While this is in line with the template of the journal, the reader cannot follow the discussion in Section 2 since the abbreviations are not explained. More about this issue is described below.  The manuscript is awkward to read due to the use of unsuitable grammar or sentence structure, or lack of meaning, for example:

(a) Although the great improvement of CO2 adsorption was achieved by amine-functionalized CNTs, however, it cannot fulfill [sic.] the ever-increasing demand of the human development."   What does it mean that the amine-functionalized CNTs achieved the "importance" of CO2 adsorption? How can a material (in this case amine-functionalized CNTs) fulfil the demand (of what?) of human development? It is difficult to capture the meaning of this sentence.

Response

We appreciate the comments of reviewer. As suggested, the related description was updated (See pages 2).

Reviewers:

(b)Five adsorption/desorption cycles are very few for sorbent materials that must sustain thousands of cycles in industrial applications, for this reason, the statement "outstanding stability with almost no decay in a successive five adsorption-desorption cycles" is misleading and wrongly placed.

Response

As suggested, this description was updated in abstract, introduction (See pages 2), and conclusion section (See pages 9).

Reviewers:

(c)Section 2, Results and Discussion: This section immediately starts with data analysis without introducing and explaining the meaning of sample name abbreviations, the reader is left wondering what  M/P-0.1-600-2 and M/P-0.1 are, for example. Sample names/abbreviations must be explained before discussing the results otherwise it is not possible to follow the paper. Note that the reader cannot wait to reach Sections 4.2 and 4.2 to finally find the required explanations. It appears that the authors moved the section "Materials and Methods" to the end of the manuscript, as required by the journal template, without assuring that the paper could be understood by the reader. 

Response

We appreciate for reviewer’s good suggestion. We added the related description for the sample name abbreviations in the result and discussion section (See pages 2).

Reviewers:

(d)Also, immediately after, the authors write about the "successful peroxidation reaction of MWCNTs", but this reaction is not described, again the reader is left with no information about how the materials were prepared. As a matter of fact, the word "peroxidation" is mentioned only once in the paper, so there is either no information about the reaction or it is not possible to find it. Later on (line 84) another sample abbreviation is introduced "M/P-0.1-T-2" but again without any previous description of what that is and how it was prepared.  

Response

We corrected the word “peroxidation” to be “Pre-oxidation”. MWCNTs was first preoxidized to be carboxylated MWCNTs under the acidic condition. The pre-oxidation process is significant for uniformly coating PANI on the surface of MWCNTs. The corresponding description of experiment section was highlighted in 3.2 fabrication of MWCNTs/PANI namocomposite (See pages 8). The detailed description of abbreviation “M/P-0.1-T-2” was in experiment section, and we also added the related description in Section 2 (See pages 2).

Reviewers:

(e) Please revise this statement:   "The fitted parameters (Table S4) shows [sic.] that LDF models possesses [sic.] a higher R2 and lower χ2 coefficient value than that of Fick, indicating that the CO2 adsorption kinetics of M/P-0.1-600-2 follows the LDF model, suggesting fast initial physical adsorption. The regeneration property is the [sic.] significant for solid adsorbents."   In Table S4 the R2 of the Fick model is better than that of LDF, the same applies for χ2. Also, the LDF fitting line does not follow the experimental data in Figure 5c, so it is not correct to state that "the CO2 adsorption kinetics of M/P-0.1-600-2 follows the LDF model". 

Response

We appreciate for reviewer’s good suggestion. As suggested, we corrected the related description (See pages 7).

Reviewers:

(f) The wording "raman spectrum" is incorrect since Raman must be capitalized. Line 215: "Then, The" double capitalized sentence. Please proofread very carefully to remove all English issues.

Response

As suggested, we corrected the mistake accordingly.

Reviewer 3 Report

In this work, the authors prepared a MWCNTs and N-rich porous carbon (derived from PANI) composite for CO2 capture. This work performed a systematic study of these MWCNT/PANI derived carbon composite by changing the M/P ratio, adjusting the amount of KOH and pyrolysis temperature. Meanwhile, the authors showed the CO2 capture performance of these composite materials. The sample M/P-0.1-600-2 showed a CO2 adsorption capacity of 2.63 mmol/g at 298K and 5 bar, which is the highest among these composite materials. However, such CO2 capture performance is not attractive comparing with literature reported results, especially with PANI derived carbons. The resulting of this work is still interesting, but a major revision should be considered before published. Here are some issues need to be addressed:

  1. The CO2 adsorption capacities in ref. 21 and 22 is way higher than the results authors reported. The materials used in ref. 21 and 22 were PANI derived N-rich carbon. Authors should provide the results of PANI (synthesized in this work) derived carbon for comparison to prove the “necessity” of introducing MWCNTs to the adsorbents.
  2. The authors failed to explain the reason that M/P-0.03-600-2 showed a lower CO2 capacity than M/P-0.1-600-2. N content for M/P-0.03-x-y and M/P-0.05-x-y samples should be provided. The authors claimed that CO2 diffusion in M/P-0.1-600-2 was better. Authors should provide adsorption kinetics of M/P-0.03-600-2 and M/P-0.05-600-2 to prove the statement.
  3. The XRD result of M/P-0.1-600-2 wasn't enough to prove the “uniform PANI coating on MWCNTs”. It can only show the existence of PANI.

Author Response

Reviewers:

In this work, the authors prepared a MWCNTs and N-rich porous carbon (derived from PANI) composite for CO2 capture. This work performed a systematic study of these MWCNT/PANI derived carbon composite by changing the M/P ratio, adjusting the amount of KOH and pyrolysis temperature. Meanwhile, the authors showed the CO2 capture performance of these composite materials. The sample M/P-0.1-600-2 showed a CO2 adsorption capacity of 2.63 mmol/g at 298K and 5 bar, which is the highest among these composite materials. However, such CO2 capture performance is not attractive comparing with literature reported results, especially with PANI derived carbons. The resulting of this work is still interesting, but a major revision should be considered before published. Here are some issues need to be addressed:

Response

We are grateful for the reviewer’s good comments. As suggested, the manuscript has been corrected accordingly.

Reviewers:

1)The CO2 adsorption capacities in ref. 21 and 22 is way higher than the results authors reported. The materials used in ref. 21 and 22 were PANI derived N-rich carbon. Authors should provide the results of PANI (synthesized in this work) derived carbon for comparison to prove the “necessity” of introducing MWCNTs to the adsorbents.

Response

We appreciate to the good suggestion. As suggested, we prepared the porous carbon derived from pure PANI. It delivered a relatively poor CO2 adsorption capacity of 1.16 mmol g-1 at 298 K and 5 bar, which further confirms the enhanced CO2 capture capability after MWCNTs addition. The corresponding value was added in Table S2. The related description was added in the manuscript (See pages 7).

Reviewers:

2) The authors failed to explain the reason that M/P-0.03-600-2 showed a lower CO2 capacity than M/P-0.1-600-2. N content for M/P-0.03-x-y and M/P-0.05-x-y samples should be provided. The authors claimed that CO2 diffusion in M/P-0.1-600-2 was better. Authors should provide adsorption kinetics of M/P-0.03-600-2 and M/P-0.05-600-2 to prove the statement.

Response

As suggested, the N content of of M/P-0.03-600-2 and M/P-0.05-600-2 was measured by EDX analysis, and the corresponding parameters were listed in Table S1. However, the value of N content is close for these three samples. This could be due to the same pyrolysis temperature. Therefore, we speculate that the MWCNTs addition is not related to the N content of actived porous carbon. As shown in Table S1, increasing the temperature of pyrolysis can decrease the N content of NPCs. M/P-0.1-600-2 showed a better performance, which could be ascribed to the development porosity. As shown in Table S2, compared with M/P-0.03-600-2 (Pore volume: 0.24 cm3 g-1; Pore size: 2.95 nm) and M/P-0.05-600-2 (Pore volume: 0.27 cm3 g-1; Pore size: 2.87 nm), M/P-0.1-600-2 presented relatively large pore volume (0.37 cm3 g-1) and smaller average pore size (2.71 nm). Therefore, we speculate that M/P-0.1-600-2 possessed a good CO2 diffusion, corresponding to its good CO2 adsorption capacity. As suggested, we also conducted the adsorption kinetics of these two samples. However, the results cannot be accurately obtained within 10 days. Therefore, we update the related description to make the sentence structure more exactly.

Reviewers:

3)The XRD result of M/P-0.1-600-2 wasn't enough to prove the “uniform PANI coating on MWCNTs”. It can only show the existence of PANI.

Response

As suggested, we updated the description of XRD (See pages 3).

Round 2

Reviewer 2 Report

The authors have addressed the reviewer's comments. I would suggest having the editorial office checking the English format.

Reviewer 3 Report

The authors have made changes accordingly. The manuscript is good to be published.